# OpenReview forum: "A2ASecBench: A Protocol-Aware Security Benchmark for Agent-to-Agent Multi-Agent Systems"
_ICLR.cc/2026/Conference — ICLR 2026 Poster_

### Official Review · Reviewer_WsnG · 2025-10-17

**Soundness:** 2
**Presentation:** 3
**Contribution:** 2
**Rating:** 4
**Confidence:** 2

**Summary:**

This paper presents A2ASECBENCH, the first protocol-aware security benchmark framework for agent-to-agent multi-agent systems (A2A-MAS), which is built on a taxonomy categorizing A2A risks into supply-chain manipulations and protocol-logic weaknesses (covering six concrete attacks) and a joint safety–utility evaluation methodology; empirical validation across travel, healthcare, and finance domains shows the identified attacks are highly effective in bypassing default safeguards, highlighting the urgent need for protocol-level defenses and standardized benchmarking.

**Strengths:**

- The paper pioneers a comprehensive threat taxonomy and model for A2A-MAS, categorizing risks into supply-chain manipulations and protocol-logic weaknesses while detailing six concrete attacks, which fills the gap of low-level, protocol-specific vulnerability exploration in existing LLM-MAS security research.
- A2ASECBENCH, the first A2A-specific security benchmark framework proposed, incorporates a dynamic adapter layer for heterogeneous deployments and a joint safety–utility evaluation methodology, enabling probing of diverse unexplored attack vectors and explicit measurement of the harmlessness-helpfulness trade-off.
- The empirical evaluation is rigorous and impactful: it tests the framework on official A2A demos across three high-stakes domains (travel, healthcare, finance), with most attacks achieving 100% success rates, clearly revealing systemic protocol-level vulnerabilities in current A2A deployments.
- The paper provides practical and actionable insights for multiple stakeholders (agent developers, system designers, protocol researchers), such as progress-aware orchestration and verifiable capability claims, and advocates layered defenses to guide the secure design and adoption of A2A ecosystems.
- Artifact is provided at the submission stage.

**Weaknesses:**

To be honest, I am not very familiar with the A2A protocol and the formal security research of multi-agent systems. I have tried my best to understand the attack design part and believe that the theoretical design of the benchmark (A2ASECBENCH) is largely sound. However, my main considerations lie in the experimental aspect, particularly regarding the generalizability of this A2A-focused benchmark:
As the authors acknowledged, the evaluation of A2ASECBENCH is limited to the official implementation of the A2A protocol. This makes me question whether the benchmark’s design is sufficiently general to bring substantial contributions to the broader multi-agent security community—for example, whether its core design ideas (such as the threat taxonomy, dynamic adapter layer, or joint safety-utility evaluation) can be extended to other multi-agent collaboration frameworks beyond A2A.
Moreover, the benchmark’s design appears to be highly coupled with the A2A protocol itself. Given that the A2A community does not seem particularly widespread yet, this coupling may restrict its application scope in practice. That said, it is worth noting that the benchmark’s design depth is relatively solid, its structured threat categorization does provide valuable insights for multi-agent security research, even if its direct applicability is currently bounded by the A2A ecosystem.

Moreover, a limitation section should be presented.

**Questions:**

Please refer to the Weaknesses section, as I am not sure about the potential impact and contribution of this work to MAS Security Community. If more explanation on this can be provided, I could consider changing my decision, though low confidence is still preserved.

---

> ### Author Response · Authors · 2025-11-27
> **Thank you, reviewer WsnG! (1/3)**
>
> ## Response for Weaknesses
>
> > I have tried my best to understand the attack design part and believe that the theoretical design of the benchmark (A2ASECBENCH) is largely sound. However, my main considerations lie in the experimental aspect, particularly regarding the generalizability of this A2A-focused benchmark: As the authors acknowledged, the evaluation of A2ASECBENCH is limited to the official implementation of the A2A protocol. This makes me question whether the benchmark’s design is sufficiently general to bring substantial contributions to the broader multi-agent security community—for example, whether its core design ideas (such as the threat taxonomy, dynamic adapter layer, or joint safety-utility evaluation) can be extended to other multi-agent collaboration frameworks beyond A2A. Moreover, the benchmark’s design appears to be highly coupled with the A2A protocol itself. Given that the A2A community does not seem particularly widespread yet, this coupling may restrict its application scope in practice. That said, it is worth noting that the benchmark’s design depth is relatively solid, its structured threat categorization does provide valuable insights for multi-agent security research, even if its direct applicability is currently bounded by the A2A ecosystem.
>
> We sincerely appreciate the reviewer’s positive assessment of our work and their thoughtful comment regarding applicability. In Table 1 and 2, we provide evidence of A2A’s growing real-world traction, across both academia and industry, as an emerging foundational infrastructure for multi-agent systems, analogous to how HTTP standardizes communication across the internet. Our work represents an initial step toward rigorous security evaluation for A2A-based ecosystems and toward developing A2AS (an analogue of HTTPS) as a trustworthy protocol foundation for secure multi-agent coordination.
>
> **Table 1.** Academic Evidence of A2A Development (Nov 2025)
> | Work | Link to Paper |
> |------|---------------|
> | AgentMaster: A Multi-Agent Conversational Framework Using A2A and MCP Protocols for Multimodal Information Retrieval and Analysis | [Paper](https://aclanthology.org/2025.emnlp-demos.5.pdf) |
> | Agent Communications toward Agentic AI at Edge -- A Case Study of the Agent2Agent Protocol | [Paper](https://arxiv.org/abs/2508.15819) |
> | Privacy in Action: Towards Realistic Privacy Mitigation and Evaluation for LLM-Powered Agents | [Paper](https://arxiv.org/abs/2509.17488) |
> | Agent-to-Agent (A2A) Protocol Integrated Digital Twin System with AgentIQ for Multimodal AI Fitness Coaching and Personalized Well-Being | [Paper](https://dl.acm.org/doi/pdf/10.1145/3746027.3758176) |
> | Anemoi: A Semi-Centralized Multi-agent System Based on Agent-to-Agent Communication MCP server from Coral Protocol | [Paper](https://arxiv.org/abs/2508.17068) |
> | Towards Engineering Multi-Agent LLMs: A Protocol-Driven Approach | [Paper](https://arxiv.org/abs/2510.12120) |
> | Which LLM Multi-Agent Protocol to Choose? | [Paper](https://arxiv.org/abs/2510.17149) |
> | Towards Multi-Agent Economies: Enhancing the A2A Protocol with Ledger-Anchored Identities and x402 Micropayments for AI Agents | [Paper](https://arxiv.org/abs/2507.19550) |
>
> **Table 2.** Industry Evidence of A2A Development (Nov 2025)
> | Vendor        | Offering                                      | Link |
> |---------------|-----------------------------------------------|------|
> | Google Cloud  | AI Agent Marketplace                          | [Product](https://cloud.google.com/blog/topics/partners/google-cloud-ai-agent-marketplace) |
> | Microsoft     | Azure AI Foundry                              | [Product](https://azure.microsoft.com/en-us/products/ai-foundry) |
> | Microsoft     | Copilot Studio                                | [Product](https://www.microsoft.com/en-us/microsoft-365-copilot/microsoft-copilot-studio) |
> | ServiceNow    | AI Agent Fabric                               | [Product](https://www.servicenow.com/now-platform/ai-agent-fabric.html) |
> | Salesforce    | AgentForce                                    | [Product](https://www.salesforce.com/ap/agentforce/) |
> | Salesforce    | AgentExchange                                 | [Product](https://agentexchange.salesforce.com/) |
> | Box           | Box AI Agents                                 | [Product](https://www.box.com/agents) |
> | UiPath        | UiPath Platform™ for Agentic Automation       | [Product](https://www.uipath.com/platform/agentic-automation) |
> | SAP           | Joule Studio                                  | [Product](https://www.sap.com/products/artificial-intelligence/joule-studio.html) |

---

> ### Author Response · Authors · 2025-11-27
> **Thank you, reviewer WsnG! (2/3)**
>
> We argue that the underlying attack patterns we study generalize beyond A2A to other ecosystems and emerging agent communication protocols. To substantiate this, we implemented two additional MAS environments based on LangGraph [1] and ANP [2], respectively.
>
> For LangGraph-based MAS, we evaluated CO, ASRF, and ATSI. We excluded AS and CC because LangGraph does not provide autonomous agent discovery or supply-chain capability registration, and we excluded HOTF because agents communicate only through the graph controller without intermediate communication states. We re-instantiate the underlying patterns: (i) For CO, we implement a LangGraph-based coordinator and craft queries that introduce inter-dependent state conditions, causing the router to repeatedly re-enqueue or re-select the same subgraph based on malicious state flags, resulting in a non-terminating loop. (ii) For ASRF, we craft malicious queries that are passed to a higher-privilege node, causing it to execute unintended actions on the attacker’s behalf.
> (iii) For ATSI, we let a node emit untrusted artifact content that is later rendered by a downstream UI or agent without sanitization.
>
> For ANP-based MAS, all six attack patterns were transferred. (i) For AS, we targeted ANP’s AgentDescription by generating nine perturbed lookalikes from one benign description and evaluating selection accuracy. (ii) For CC, we injected an agent whose claimed and actual capabilities diverged, and ANP’s peer network failed to detect the mismatch. (iii) For HOTF, we stalled one agent in an intermediate meta-protocol negotiation state, causing the negotiation to never terminate. (iv) For CO, ASRF, and ATSI, we delivered the corresponding payloads through ANP’s peer-communication examples and found they remained effective, as ANP propagated the payloads end-to-end without detection or sanitization.
>
> The results are shown in Table 3 and 4. From the result, we can observe that our attack patterns can successfully transfer to MAS built on LangGraph and ANP with mostly 100% ASR. These results verify that although our main focus is the A2A, our discovered attack patterns can transfer to other MAS baselines.
>
> **Table 3.**  Transfer to LangGraph
> | Attack Pattern                   | ASR |
> |----------------------------------|-----------|
> | Cycle Overflow                  | 100%         |
> | Agent-Side Request Forgery      | 100%         |
> | Artifact-Triggered Script Injection | 100%     |
>
> **Table 4.**  Transfer to ANP
> | Attack Pattern                   | ASR  |
> |----------------------------------|------|
> | AgentCard Spoofing              | 98% |
> | Capability Cloaking             | 100%    |
> | Half-Open Task Flooding         | 100%    |
> | Cycle Overflow                  | 100%    |
> | Agent-Side Request Forgery      | 100%  |
> | Artifact-Triggered Script Injection | 100% |
>
> [1] https://www.langchain.com/langgraph
> [2] Chang, G., Lin, E., Yuan, C., Cai, R., Chen, B., Xie, X., & Zhang, Y. (2025). Agent network protocol technical white paper. arXiv preprint arXiv:2508.00007.
>
>
>
>
> > Moreover, a limitation section should be presented.
>
> Thank you for the suggestion. We have added a dedicated Limitations section. Our work primarily focuses on identifying threat patterns and constructing a benchmark intended to help the community design more secure MAS prototypes. While we evaluate several basic defenses with Nvidia NeMo Guardrail (see Table 5 and analysis below) and outline potential mitigation directions, we do not propose a fully secure MAS design. Developing a robust, security-hardened prototype remains an important direction for future work.
>
> Beyond the host-prompt hardening discussed in the paper, we further integrated NVIDIA NeMo Guardrails [3], one of the most mature, production-oriented guardrails, as a security gateway. Table 5 shows that such guardrails offer only limited protection against our proposed MAS-specific attacks. HOTF and ATSI still succeed at high rates (≥0.85 and ≥0.91), CO is only partially suppressed (0.66–0.73), and even ASRF, the most mitigated because of obvious pattern like sensitive internal uri, retains non-trivial success (0.23–0.48). These results reveal a fundamental gap: existing guardrails are not designed to understand multi-agent interaction patterns, state transitions, or protocol semantics, and thus cannot reliably defend against MAS-specific misuse.
>
> **Table 5.**  Performance against guardrail
> | Attack with Guardrail         | Travel | Healthcare | Finance |
> |----------------------------------|--------|------------|---------|
> | Half-Open Task Flooding          | 91%   | 85%       | 90%    |
> | Cycle Overflow                   | 66%   | 73%       | 70%    |
> | Agent-Side Request Forgery       | 37%   | 23%       | 48%    |
> | Artifact-Triggered Script Injection | 94% | 93%       | 91%    |

---

> ### Author Response · Authors · 2025-11-27
> **Thank you, reviewer WsnG! (3/3)**
>
> ## Response for Questions
>
> > Please refer to the Weaknesses section, as I am not sure about the potential impact and contribution of this work to MAS Security Community. If more explanation on this can be provided, I could consider changing my decision, though low confidence is still preserved.
>
> We appreciate the reviewer’s concern and have clarified the contribution and impact more explicitly. To further articulate the potential impact of this work, we added an Impact Statement section that highlights the following points. (i) First, A2ASecBench provides the first protocol-aware security benchmark for A2A multi-agent systems, transforming previously informal concerns about MAS misuse into measurable, reproducible evaluation criteria. (ii) Second, the benchmark introduces a structured threat taxonomy and a reusable dynamic adapter design that can guide future research on both A2A and emerging agent communication protocols. (iii) Third, by demonstrating that all six attack patterns transfer to LangGraph- and ANP-based MAS, our work reveals underlying vulnerability patterns that generalize beyond any single protocol, informing the broader MAS security community. (iv) Fourth, the defense results uncover a gap between today’s guardrails and MAS-specific threats, motivating research into protocol-level hardening, secure registries, and MAS-native defenses. (v) Finally, A2ASecBench serves as an early foundation for developing secure standards, providing a basis for future work on threat modeling, defense evaluation, and trustworthy agent interoperability.

---

### Official Review · Reviewer_gs3A · 2025-10-30

**Soundness:** 3
**Presentation:** 3
**Contribution:** 3
**Rating:** 8
**Confidence:** 3

**Summary:**

This paper fills the gap of lacking protocol-specific security benchmarks for A2A-MAS. It proposes A2ASECBENCH, a framework with a threat taxonomy (6 attacks), scenario adapter, and joint safety-utility evaluation. Key results: 5 attacks achieve 100% ASR across travel/healthcare/finance, highlighting A2A’s systemic vulnerabilities.

**Strengths:**

1. First A2A-specific benchmark: It covers 6 novel attack vectors (e.g., Cycle Overflow, ATSI) spanning supply-chain and protocol-logic risks, unlike generic LLM benchmarks, enabling targeted A2A security evaluation.
2. Scenario adaptability: The dynamic adapter maps attacks to heterogeneous A2A stacks (e.g., travel/healthcare), ensuring portability—e.g., it generates domain-specific test cases by aligning attacks with scenario specs.
3. Joint safety-utility evaluation: It pairs adversarial trials with benign tasks to measure trade-offs (e.g., Capability Cloaking reduces travel utility from 0.853 to 0.682), avoiding one-sided safety assessment.

**Weaknesses:**

1. Limited defense testing: It focuses on attack effectiveness but only tests host prompt hardening as mitigation; other defenses (e.g., protocol-level signatures, peer authentication) are unexamined—adding diverse defenses would improve benchmark comprehensiveness.
2. Small-scale MAS testing: It evaluates MAS with 1 host + 3 remote agents; large-scale MAS (10+ agents) are untested—testing larger systems would confirm scalability of vulnerabilities.
3. Lack of production deployment validation: It uses official A2A demos but not real-world production stacks; testing on deployed systems would enhance practical relevance.

**Questions:**

Please refer to the weaknesses above.

---

> ### Author Response · Authors · 2025-11-27
> **Thank you, reviewer gs3A! (1/2)**
>
> ## Response for Weaknesses
> > Limited defense testing: It focuses on attack effectiveness but only tests host prompt hardening as mitigation; other defenses (e.g., protocol-level signatures, peer authentication) are unexamined—adding diverse defenses would improve benchmark comprehensiveness.
>
> We appreciate the reviewer’s insight. Our work focuses primarily on characterizing attack effectiveness, and we agree that broader defense evaluation is valuable. Beyond the host-prompt hardening discussed in the paper, we further integrated NVIDIA NeMo Guardrails [3], one of the most mature, production-oriented guardrails, as a security gateway. Table 1 shows that such guardrails offer only limited protection against our proposed MAS-specific attacks. HOTF and ATSI still succeed at high rates (≥0.85 and ≥0.91), CO is only partially suppressed (0.66–0.73), and even ASRF, the most mitigated because of obvious pattern like sensitive internal uri, retains non-trivial success (0.23–0.48). These results reveal a fundamental gap: existing guardrails are not designed to understand multi-agent interaction patterns, state transitions, or protocol semantics, and thus cannot reliably defend against MAS-specific misuse.
>
> **Table 1.**  Performance against guardrail
> | Attack with Guardrail         | Travel | Healthcare | Finance |
> |----------------------------------|--------|------------|---------|
> | Half-Open Task Flooding          | 91%   | 85%       | 90%    |
> | Cycle Overflow                   | 66%   | 73%       | 70%    |
> | Agent-Side Request Forgery       | 37%   | 23%       | 48%    |
> | Artifact-Triggered Script Injection | 94% | 93%       | 91%    |
>
> While defenses such as peer authentication, signed AgentCards, and capability attestation can reduce some risks (e.g., spoofing), they cannot address the core vulnerabilities exposed by our benchmark. Even fully authenticated agents can still trigger HOTF, CO, ASRF, or ATSI, because these attacks arise from protocol semantics, multi-step state transitions, and artifact rendering rather than identity forgery. Moreover, in realistic MAS deployments, universal authentication is impractical, open ecosystems rarely achieve complete certification, and even mature markets with strong signing infrastructures (e.g., VS Code [2], npm [3]) still suffer lookalike abuses. Thus, identity- or signature-based defenses help but are inherently limited, and prompt-level guardrails fail to address protocol-rooted failure modes. Our work therefore highlights structural weaknesses that persist under realistic defenses.
>
> [1] Rebedea, T., Dinu, R., Sreedhar, M. N., Parisien, C., & Cohen, J. (2023, December). Nemo guardrails: A toolkit for controllable and safe llm applications with programmable rails. In Proceedings of the 2023 conference on empirical methods in natural language processing: system demonstrations (pp. 431-445).
>
> [2] Edirimannage, S., Elvitigala, C., Don, A. K. K., Daluwatta, W., Wijesekara, P., & Khalil, I. (2024). Developers Are Victims Too: A Comprehensive Analysis of The VS Code Extension Ecosystem. arXiv preprint arXiv:2411.07479.
>
> [3] https://blog.phylum.io/malicious-packages-typosquatting-and-other-attacks-against-open-source-dependencies

---

> ### Author Response · Authors · 2025-11-27
> **Thank you, reviewer gs3A! (2/2)**
>
> > Small-scale MAS testing: It evaluates MAS with 1 host + 3 remote agents; large-scale MAS (10+ agents) are untested—testing larger systems would confirm scalability of vulnerabilities.
>
> We appreciate the reviewer’s point regarding larger MAS configurations. Our evaluation focuses on protocol-level vulnerabilities, which are triggered by A2A interactions between one client agent and one remote agent, and do not require many agents to manifest. As shown in our threat model and system design, we exploit the A2A protocol’s discovery, orchestration, or artifact-exchange semantics, making a 1-host + 3-agent setup sufficient to reveal protocol-rooted weaknesses. However, we believe that large-scale MAS evaluation remains highly valuable for peer-to-peer (P2P) agent networks by ANP [1], in contrast to A2A’s client-server (CS) model, and we are running additional large-scale MAS experiments to reinforce our evaluation. These take time and we will follow up with the results once they are ready.
>
> [1] Chang, G., Lin, E., Yuan, C., Cai, R., Chen, B., Xie, X., & Zhang, Y. (2025). Agent network protocol technical white paper. arXiv preprint arXiv:2508.00007.
>
> > Lack of production deployment validation: It uses official A2A demos but not real-world production stacks; testing on deployed systems would enhance practical relevance.
>
> We appreciate the reviewer’s concern regarding production deployment validation. Because the A2A ecosystem is still in its early stage,to the best of our knowledge there are currently no publicly available, production-grade A2A systems that would allow third-party security evaluation.The official A2A implementation released by Google is therefore the most realistic and representative deployment environment available today. Evaluating against this reference implementation is appropriate because it defines the protocol semantics and interaction patterns that downstream systems are expected to follow as the ecosystem matures.
>
> Importantly, the threats we identify, such as AS/CC/HOTF/ASRF/etc., stem from structural properties of the A2A protocol and its interaction model, rather than quirks of a particular demo. Because production deployments are expected to reuse the same protocol constructs (e.g., AgentCards, artifact exchange), these vulnerabilities are not demo-specific, but generalizable to any future A2A-based system unless mitigated.
>
> Finally, from a security research perspective, proactively analyzing the reference implementation at this early stage is valuable: it allows the community to address systemic weaknesses before production deployments emerge, substantially reducing long-term risk and remediation cost. This aligns with established practice in security for emerging standards, where early threat modeling and proof-of-concept evaluation are performed before real-world systems are widely deployed.

---

### Official Review · Reviewer_sNRg · 2025-11-01

**Soundness:** 3
**Presentation:** 3
**Contribution:** 3
**Rating:** 4
**Confidence:** 3

**Summary:**

This paper presents a benchmark framework for studying the multi-agent security of the Agent-to-Agent protocol, named A2ASecBENCH. Using A2ASecBENCH, this paper identifies six types of security threats present in the A2A protocol.

**Strengths:**

S1. This work is well-motivated, well-structured, and clearly presented.

S2. The proposed benchmark framework is rigorously defined mathematically.

S3.  A2ASecBENCH reveals six critical types of attacks within the A2A ecosystem, which are essential for comprehensively probing security risks in this ecosystem.

S4. The authors validate the effectiveness of the six identified attack types in three representative A2A protocol application scenarios.

S5. Potential defenses are discussed.

**Weaknesses:**

W1. The applicability of A2ASecBENCH is limited. Although I consider its design to be solid, it targets only the A2A protocol proposed by Google.

W2. The following two papers also systematically analyze security vulnerabilities in the A2A protocol; however, this paper does not thoroughly discuss their similarities and differences.

[1] Building a secure agentic AI application leveraging A2A protocol.

[2] Improving Google A2A Protocol: Protecting Sensitive Data and Mitigating Unintended Harms in Multi-Agent Systems.

**Questions:**

Q1. I am curious about the generality of the six types of attacks identified by A2ASecBENCH and whether they also exist in other multi-agent communication protocols.

---

> ### Author Response · Authors · 2025-11-27
> **Thank you, reviewer sNRg! (1/2)**
>
> ## Response for Weaknesses
> > The applicability of A2ASecBENCH is limited. Although I consider its design to be solid, it targets only the A2A protocol proposed by Google.
>
> Thank you for the positive assessment of our benchmark design. We acknowledge that A2ASecBench is currently scoped to the A2A protocol. Our motivation is that A2A is a promising candidate for foundational multi-agent infrastructure, analogous to how HTTP standardizes client-server communication, and thus warrants early, protocol-aware security evaluation. We show evidence of A2A’s rapid development and real-world impact for both industry and academic by the end of Introduction.
>
> Furthermore, we agree that demonstrating applicability is essential. To this end, we implemented two additional MAS variants, one built on LangGraph [1] and another on ANP [2].
>
> For LangGraph-based MAS, we evaluated CO, ASRF, and ATSI. We excluded AS and CC because LangGraph does not provide autonomous agent discovery or supply-chain capability registration, and we excluded HOTF because agents communicate only through the graph controller without intermediate communication states. We re-instantiate the underlying patterns: (i) For CO, we implement a LangGraph-based coordinator and craft queries that introduce inter-dependent state conditions, causing the router to repeatedly re-enqueue or re-select the same subgraph based on malicious state flags, resulting in a non-terminating loop. (ii) For ASRF, we craft malicious queries that are passed to a higher-privilege node, causing it to execute unintended actions on the attacker’s behalf.
> (iii) For ATSI, we let a node emit untrusted artifact content that is later rendered by a downstream UI or agent without sanitization.
>
> For ANP-based MAS, all six attack patterns were transferred. (i) For AS, we targeted ANP’s AgentDescription by generating nine perturbed lookalikes from one benign description and evaluating selection accuracy. (ii) For CC, we injected an agent whose claimed and actual capabilities diverged, and ANP’s peer network failed to detect the mismatch. (iii) For HOTF, we stalled one agent in an intermediate meta-protocol negotiation state, causing the negotiation to never terminate. (iv) For CO, ASRF, and ATSI, we delivered the corresponding payloads through ANP’s peer-communication examples and found they remained effective, as ANP propagated the payloads end-to-end without detection or sanitization.
>
> The results are shown in Table 1 and 2. From the result, we can observe that our attack patterns can successfully transfer to MAS built on LangGraph and ANP with mostly 100% ASR. These results verify that although our main focus is the A2A, our discovered attack patterns can transfer to other MAS baselines.
>
> **Table 1.**  Transfer to LangGraph
> | Attack Pattern                   | ASR |
> |----------------------------------|-----------|
> | Cycle Overflow                  | 100%         |
> | Agent-Side Request Forgery      | 100%         |
> | Artifact-Triggered Script Injection | 100%     |
>
> **Table 2.**  Transfer to ANP
> | Attack Pattern                   | ASR  |
> |----------------------------------|------|
> | AgentCard Spoofing              | 98% |
> | Capability Cloaking             | 100%    |
> | Half-Open Task Flooding         | 100%    |
> | Cycle Overflow                  | 100%    |
> | Agent-Side Request Forgery      | 100%  |
> | Artifact-Triggered Script Injection | 100% |
>
> [1] https://www.langchain.com/langgraph
>
> [2] Chang, G., Lin, E., Yuan, C., Cai, R., Chen, B., Xie, X., & Zhang, Y. (2025). Agent network protocol technical white paper. arXiv preprint arXiv:2508.00007.
>
> > The following two papers also systematically analyze security vulnerabilities in the A2A protocol; however, this paper does not thoroughly discuss their similarities and differences.
> > [1] Building a secure agentic AI application leveraging A2A protocol.
> > [2] Improving Google A2A Protocol: Protecting Sensitive Data and Mitigating Unintended Harms in Multi-Agent Systems.
>
> We appreciate the reviewer for pointing out these works. We have expanded the Related Work section to compare our contributions with [1] and [2]. Both papers provide qualitative analyses and security recommendations, but neither introduces formalized A2A-specific attack vectors nor an executable adversarial benchmark. In contrast, our work contributes the first protocol-aware, end-to-end evaluation framework that systematically implements and measures six attack vectors across the full A2A lifecycle. We have clarified these distinctions in the revision.
>
> [1] Habler, I., Huang, K., Narajala, V. S., & Kulkarni, P. (2025). Building a secure agentic AI application leveraging A2A protocol. arXiv preprint arXiv:2504.16902.
>
> [2] Louck, Y., Stulman, A., & Dvir, A. (2025). Improving Google A2A Protocol: Protecting Sensitive Data and Mitigating Unintended Harms in Multi-Agent Systems. arXiv preprint arXiv:2505.12490.

---

> ### Author Response · Authors · 2025-11-27
> **Thank you, reviewer sNRg! (2/2)**
>
> ## Response for Questions
> > I am curious about the generality of the six types of attacks identified by A2ASecBENCH and whether they also exist in other multi-agent communication protocols.
>
> Thank you for the thoughtful question. To assess generality, we extended our evaluation to two representative non-A2A ecosystems, LangGraph [1] and ANP [2], and implemented MAS instances in both. As per previous response (1/2), the underlying attack patterns do transfer, indicating that these vulnerabilities are not unique to A2A but reflect broader structural weaknesses shared across multi-agent communication protocols.
>
> [1] https://www.langchain.com/langgraph
>
> [2] Chang, G., Lin, E., Yuan, C., Cai, R., Chen, B., Xie, X., & Zhang, Y. (2025). Agent network protocol technical white paper. arXiv preprint arXiv:2508.00007.

---

### Official Review · Reviewer_wjZT · 2025-11-03

**Soundness:** 3
**Presentation:** 3
**Contribution:** 3
**Rating:** 6
**Confidence:** 3

**Summary:**

The paper shows that multi-agent LLM systems using the A2A protocol have a protocol-level attack surface that prompt-level safety can’t see. It introduces A2ASecBench, a protocol-aware benchmark that encodes six concrete attacks (two supply-chain, four protocol-logic) across travel, healthcare, and finance, and evaluates both attack success and benign-task utility. Running it on real A2A demos, five of six attacks succeed nearly 100% of the time, and spoofed/capability-cloaked agents can hijack or degrade normal workflows, proving the registry’s trust model is too weak.

**Strengths:**

1. Propose the first protocol-aware benchmark (A2ASecBench) with six instantiated attacks on three domains (travel, healthcare, finance)
2. Strong empirical evidence of systemic vulnerability. On official A2A samples, five of six attacks hit 100% ASR across all domains, and even the “harder” AgentCard Spoofing still succeeds ~0.82–0.83.
3. Clear and easy to understand writing with takeaways

**Weaknesses:**

1. Limited empirical scope (only A2A). All results are on “official A2A samples” in three domains; there’s no evidence the attacks transfer to independently built, messier A2A stacks or to agent platforms that already add extra validation. This weakens the “protocol-wide” claim. Adding at least one non-official, differently engineered MAS baseline (e.g., with custom task gating or artifact sanitization) would make the evaluation harder to dismiss.
2. Assumptions about rendering/consumption in ATSI. The Artifact-Triggered Script Injection attack inherits XSS-like power only if the host or a downstream agent renders artifacts in a permissive way; the paper does not enumerate which of the standard A2A sample apps actually do that. A tighter analysis of artifact types vs. vulnerable renderers would help readers know when ATSI is real and when it’s theoretical.
3. Significance tied to A2A’s adoption curve. The paper’s impact claim leans on A2A being the interoperability layer for agents, but the experiments don’t show applicability to adjacent ecosystems (OpenAI’s agent runtime, LangGraph-style planners, or in-house orchestrators). Porting 1–2 attacks through the “dynamic adapter” into a non-A2A stack would make the contribution less protocol-fragile.

**Questions:**

1. how many concurrent tasks, for how long, and on what hardware/config are used in DoS like attacks ? Adding resource-usage curves (tasks vs. latency/queue depth) would make the threat more operational and results more clear.
2. Would smarter registries (e.g., fuzzy matching, issuer-based trust, or signed manifests) help in the AgentCard Spoofing attack? How many lookalike cards are needed to get selected with high probability? A short sensitivity study on discovery ranking would turn this from a single demo into a reusable security test.

**Details Of Ethics Concerns:**

This paper implements several attacks to invade multi-agent system and A2A protocol, which could be potentially leveraged by malicious attackers in realworld.

---

> ### Author Response · Authors · 2025-11-27
> **Thank you, reviewer wjZT! (1/4)**
>
> ## Response for Weaknesses
> > Limited empirical scope (only A2A). All results are on “official A2A samples” in three domains; there’s no evidence the attacks transfer to independently built, messier A2A stacks or to agent platforms that already add extra validation. This weakens the “protocol-wide” claim. Adding at least one non-official, differently engineered MAS baseline (e.g., with custom task gating or artifact sanitization) would make the evaluation harder to dismiss.
>
> We thank the reviewer for raising this important point. Following your suggestion, we implemented two additional MAS, one built on LangGraph [1] and another on ANP [2].
>
> For LangGraph-based MAS, we evaluated CO, ASRF, and ATSI. We excluded AS and CC because LangGraph does not provide autonomous agent discovery or supply-chain capability registration, and we excluded HOTF because agents communicate only through the graph controller without intermediate communication states. We re-instantiate the underlying patterns: (i) For CO, we implement a LangGraph-based coordinator and craft queries that introduce inter-dependent state conditions, causing the router to repeatedly re-enqueue or re-select the same subgraph based on malicious state flags, resulting in a non-terminating loop. (ii) For ASRF, we craft malicious queries that are passed to a higher-privilege node, causing it to execute unintended actions on the attacker’s behalf.
> (iii) For ATSI, we let a node emit untrusted artifact content that is later rendered by a downstream UI or agent without sanitization.
>
> For ANP-based MAS, all six attack patterns were transferred. (i) For AS, we targeted ANP’s AgentDescription by generating nine perturbed lookalikes from one benign description and evaluating selection accuracy. (ii) For CC, we injected an agent whose claimed and actual capabilities diverged, and ANP’s peer network failed to detect the mismatch. (iii) For HOTF, we stalled one agent in an intermediate meta-protocol negotiation state, causing the negotiation to never terminate. (iv) For CO, ASRF, and ATSI, we delivered the corresponding payloads through ANP’s peer-communication examples and found they remained effective, as ANP propagated the payloads end-to-end without detection or sanitization.
>
> The results are shown in Table 1 and 2. From the result, we can observe that our attack patterns can successfully transfer to MAS built on LangGraph and ANP with mostly 100% ASR. These results verify that although our main focus is the A2A, our discovered attack patterns can transfer to other MAS baselines.
>
> **Table 1.**  Transfer to LangGraph
> | Attack Pattern                   | ASR |
> |----------------------------------|-----------|
> | Cycle Overflow                  | 100%         |
> | Agent-Side Request Forgery      | 100%         |
> | Artifact-Triggered Script Injection | 100%     |
>
> **Table 2.**  Transfer to ANP
> | Attack Pattern                   | ASR  |
> |----------------------------------|------|
> | AgentCard Spoofing              | 98% |
> | Capability Cloaking             | 100%    |
> | Half-Open Task Flooding         | 100%    |
> | Cycle Overflow                  | 100%    |
> | Agent-Side Request Forgery      | 100%  |
> | Artifact-Triggered Script Injection | 100% |
>
> We further integrated NVIDIA NeMo Guardrails [3], one of the most mature, production-oriented guardrails, as a security gateway. Table 3 shows that such guardrails offer only limited protection against our proposed MAS-specific attacks. HOTF and ATSI still succeed at high rates (≥0.85 and ≥0.91), CO is only partially suppressed (0.66–0.73), and even ASRF, the most mitigated because of obvious pattern like sensitive internal uri, retains non-trivial success (0.23–0.48). These results reveal a fundamental gap: existing guardrails are not designed to understand multi-agent interaction patterns, state transitions, or protocol semantics, and thus cannot reliably defend against MAS-specific misuse.
>
> **Table 3.**  Performance against guardrail
> | Attack with Guardrail         | Travel | Healthcare | Finance |
> |----------------------------------|--------|------------|---------|
> | Half-Open Task Flooding          | 91%   | 85%       | 90%    |
> | Cycle Overflow                   | 66%   | 73%       | 70%    |
> | Agent-Side Request Forgery       | 37%   | 23%       | 48%    |
> | Artifact-Triggered Script Injection | 94% | 93%       | 91%    |
>
> [1] https://www.langchain.com/langgraph
>
> [2] Chang, G., Lin, E., Yuan, C., Cai, R., Chen, B., Xie, X., & Zhang, Y. (2025). Agent network protocol technical white paper. arXiv preprint arXiv:2508.00007.
>
> [3] Rebedea, T., Dinu, R., Sreedhar, M. N., Parisien, C., & Cohen, J. (2023, December). Nemo guardrails: A toolkit for controllable and safe llm applications with programmable rails. In Proceedings of the 2023 conference on empirical methods in natural language processing: system demonstrations (pp. 431-445).

---

> ### Author Response · Authors · 2025-11-27
> **Thank you, reviewer wjZT! (2/4)**
>
> > Assumptions about rendering/consumption in ATSI. The Artifact-Triggered Script Injection attack inherits XSS-like power only if the host or a downstream agent renders artifacts in a permissive way; the paper does not enumerate which of the standard A2A sample apps actually do that. A tighter analysis of artifact types vs. vulnerable renderers would help readers know when ATSI is real and when it’s theoretical.
>
> We appreciate the reviewer’s insightful comment. A central reason ATSI remains a practical threat in current A2A ecosystems is that the A2A protocol does not specify any rendering or sanitization policies. By design, A2A leaves artifact handling entirely to downstream applications, meaning the protocol neither prevents permissive rendering nor guarantees safe rendering. This design choice creates an implicit attack surface: once an artifact is displayed by a renderer that interprets rich text or markup, adversarial content can acquire script-execution capability.
>
> To make this more explicit, we analyze artifact types versus vulnerable renderer behaviors. Certain artifact formats frequently produced by agents, such as HTML, Markdown with HTML passthrough, can embed executable constructs, including `<script>` tags, JavaScript code, or event-triggered handlers. These formats pose ATSI risk only when a renderer interprets them as markup instead of plain text. Vulnerable renderers include browser-based frontend and HTML engines. We also provide concrete evidence that ATSI is not hypothetical. Several sample applications use permissive browser-facing interfaces that render agent outputs as HTML without sanitization. In our end-to-end proof-of-concept demonstrated in the anonymous link below, an injected artifact containing a benign alert() payload executes immediately when displayed in the client’s browser UI. This demonstrates that ATSI can manifest as an XSS-equivalent exploit in real environments, not merely in contrived settings.
>
> [https://anonymous.4open.science/r/img-D57E/atsi_real-world_poc.png](https://anonymous.4open.science/r/img-D57E/atsi_real-world_poc.png)
>
> Together, this tighter analysis clarifies when ATSI becomes an operational concern: the risk emerges from the interaction between script-capable artifact types and permissive renderers, a scenario enabled, and currently unregulated, by the A2A design philosophy.
>
> > Significance tied to A2A’s adoption curve. The paper’s impact claim leans on A2A being the interoperability layer for agents, but the experiments don’t show applicability to adjacent ecosystems (OpenAI’s agent runtime, LangGraph-style planners, or in-house orchestrators). Porting 1–2 attacks through the “dynamic adapter” into a non-A2A stack would make the contribution less protocol-fragile.
>
> We appreciate the reviewer’s insightful comment. Following your suggestion, we extended our evaluation to two representative non-A2A ecosystems, LangGraph [1] and ANP [2], and implemented MAS instances in both frameworks. The detailed results are shown in previous response (1/4). The new experiments show that our attack patterns do transfer with adaptation, despite differences in protocol structure and system abstractions. These findings indicate that the vulnerabilities we identify stem from general agent-interaction patterns rather than limited to A2A, supporting our claim that the contribution is not protocol-fragile.
>
> [1] https://www.langchain.com/langgraph
>
> [2] Chang, G., Lin, E., Yuan, C., Cai, R., Chen, B., Xie, X., & Zhang, Y. (2025). Agent network protocol technical white paper. arXiv preprint arXiv:2508.00007.

---

> ### Author Response · Authors · 2025-11-27
> **Thank you, reviewer wjZT! (3/4)**
>
> ## Response for Questions
> > how many concurrent tasks, for how long, and on what hardware/config are used in DoS like attacks ? Adding resource-usage curves (tasks vs. latency/queue depth) would make the threat more operational and results more clear.
>
> We appreciate the reviewer’s suggestion. In the anonymous link provided below, we include resource-usage curves for the HOTF attack. As shown in the left subfigure, the number of active tasks quickly reaches the system’s upper bound of 100 concurrent tasks. Once this limit is saturated, the system enters a denial-of-service state, evidenced by the steeply increasing slope in the right subfigure.
>
> [https://anonymous.4open.science/r/img-D57E/hotf_curve.png](https://anonymous.4open.science/r/img-D57E/hotf_curve.png)
>
> While resource-usage curves can indeed aid operational tuning, our focus is on the protocol-level failure modes that fundamentally cause unbounded task growth. As summarized in Table 4, our experiments were conducted on a standard, modest workstation with a controlled ASGI server configuration.
>
> **Table 4.** Experimental Hardware and Server Configuration
> | Category | Configuration |
> |---------|---------------|
> | **Hardware** | AMD Ryzen 9 8945HX (16C/32T, 2.50 GHz), 16 GB DDR5 RAM, 64-bit OS (x64 architecture) |
> | **Application Framework** | A2A (Agent-to-Agent) protocol implementation on Python 3.13+ |
> | **Web Server** | Uvicorn ASGI server with Starlette framework |
> | **Max Concurrent Tasks** | 100 |
> | **API Framework** | FastAPI + Uvicorn (ASGI) |
>
> Even under this relatively constrained setup, once a task enters an endless intermediate state in HOTF or a deadlock state in CO, the number of active task instances increases monotonically and never resolves. This demonstrates that the DoS effect is not a consequence of hardware limitations or server throughput, but of protocol semantics themselves. We therefore argue that MAS DoS resilience should be evaluated by addressing these underlying failure modes rather than by scaling hardware or adjusting deployment configurations.

---

> ### Author Response · Authors · 2025-11-27
> **Thank you, reviewer wjZT! (4/4)**
>
> > Would smarter registries (e.g., fuzzy matching, issuer-based trust, or signed manifests) help in the AgentCard Spoofing attack? How many lookalike cards are needed to get selected with high probability? A short sensitivity study on discovery ranking would turn this from a single demo into a reusable security test.
>
> Thank you for the insightful comment. Smarter registries can certainly reduce spoofing risk but cannot eliminate the underlying issue. In the early stage of A2A, many deployments lack mature or secure registries, making AgentCard Spoofing directly applicable. Even when registries become more mature, using mechanisms such as signed manifests, issuer-based trust, or fuzzy matching, real ecosystems still suffer from lookalike abuse (e.g., the VS Code plugin ecosystem [1]), demonstrating that vulnerabilities persist even under stronger vetting. Fundamentally, it is unrealistic for every agent in a large ecosystem to be formally or officially certified, meaning spoofing surfaces inevitably reappear as the ecosystem grows. Thus, registry hardening can mitigate but cannot fully remove the structural susceptibility of discovery-based systems to lookalike attacks.
>
> Per your suggestion, we added a sensitivity study to examine how discovery ranking deteriorates as the number of injected lookalike cards (k value) increases. Table 5 shows a strong monotonic trend: as kkk grows from 3->5->7, the ASR escalates from 76%->87%->95%, indicating that even modest increases in spoofed entries cause the discovery mechanism to fail almost deterministically. This pattern is reinforced by Tables 6 and 7, which evaluate the k=10 setting used in our main experiments: in A2A, spoofed variants achieve 99% Top-1 and 87% Top-3 selection; in ANP, the attack yields 100% Top-1 and 81.25% Top-3 selection. Together, these results show that high-probability selection requires only a small number of lookalikes, and once k reaches the typical evaluation size (e.g., 10), the ranking pipeline collapses almost completely. This transforms the attack from a single demonstration into a reusable test: red teamer can vary k and directly quantify robustness, revealing the urgent need for stronger registry validation, similarity-aware filtering, and signed manifests, etc..
>
> **Table 5.** Sensitivity Study for k value in AS
> | k   | 3    | 5    | 7    |
> |-----|------|------|------|
> | ASR | 76%  | 87%  | 95%  |
>
> **Table 6.** Sensitivity Study for A2A (k=10)
> | AgentCard | Top-1 | Top-3 | Top-5 |
> |--------|-------|--------|--------|
> | ASR  | 99%    | 87% | 38% |
>
> **Table 7.** Sensitivity Study for ANP (k=10)
> | AgentDescription | Top-1 | Top-3 | Top-5 |
> |--------|-------|--------|--------|
> | ASR | 100%    | 81.25% | 18.75% |
>
> [1] Edirimannage, S., Elvitigala, C., Don, A. K. K., Daluwatta, W., Wijesekara, P., & Khalil, I. (2024). Developers Are Victims Too: A Comprehensive Analysis of The VS Code Extension Ecosystem. arXiv preprint arXiv:2411.07479.

---

### Meta-Review · Area_Chair_h7cr · 2026-01-06

**Summary:**

The reviewers generally find this paper timely, well-motivated, and technically sound, highlighting its novelty as the first protocol-aware security benchmark specifically targeting A2A-based multi-agent systems. The main strengths emphasized across reviews include the clear threat taxonomy, the concrete instantiation of six protocol-level attacks, and strong empirical evidence showing systemic vulnerabilities in official A2A demos. The primary concerns focus on limited scope (initially restricted to A2A), questions about generalization to other multi-agent frameworks, and the need for stronger operational grounding and defense evaluation.

**Reviewer Concerns:**

In the rebuttal, the authors substantially addressed the major concerns. The generalization issue was directly mitigated by new experiments on LangGraph and ANP, demonstrating that several attack patterns transfer beyond A2A, which meaningfully strengthens the paper’s relevance to the broader MAS community. Concerns about ATSI assumptions, DoS operational realism, spoofing dynamics, and limited defense evaluation were also concretely addressed through added PoCs, resource-usage curves, sensitivity studies, and experiments with NVIDIA NeMo Guardrails. While some reviewers still note that large-scale MAS deployments and broader production validation remain open, the core technical and empirical weaknesses raised in the initial reviews have been largely resolved.

**Reviewer Scores:**

I expect the initially positive reviewers to remain at similar scores. For the more critical reviewers who raised generalization and defense-related concerns (the original 4-score reviewers), the added cross-stack experiments and expanded evaluations would likely move their assessments upward, closer to the acceptance range. Overall, the paper now appears solidly within the acceptance band.

---

### Decision · Program_Chairs · 2026-01-26

Accept (Poster)